# Psychiatric Illness or Immune Dysfunction—Brain Perfusion Imaging Providing the Answer in a Case of Anti-NMDAR Encephalitis

**DOI:** 10.3390/diagnostics12102377

**Published:** 2022-09-30

**Authors:** Ines Šiško Markoš, Ivan Blažeković, Vjekoslav Peitl, Tomislav Jukić, Višnja Supanc, Dalibor Karlović, Ana Fröbe

**Affiliations:** 1Department of Oncology and Nuclear Medicine, Sestre Milosrdnice University Hospital Center, 10000 Zagreb, Croatia; 2School of Medicine, Catholic University of Croatia, 10000 Zagreb, Croatia; 3Department of Psychiatry, Sestre Milosrdnice University Hospital Center, 10000 Zagreb, Croatia; 4School of Medicine, University of Zagreb, 10000 Zagreb, Croatia; 5Department of Neurology, Sestre Milosrdnice University Hospital Center, 10000 Zagreb, Croatia; 6School of Dental Medicine, University of Zagreb, 10000 Zagreb, Croatia

**Keywords:** anti-NMDAR encephalitis, SPECT, quantification, schizophrenia

## Abstract

Background: We investigated the potential use of SPECT quantification in addition to qualitative brain perfusion analysis for the detection of anti-NMDAR encephalitis. The question is how to normalize brain activity to be able to quantitatively detect perfusion patterns. Usually, brain activity is normalized to a structure considered unaffected by the disease. Methods: Brain [99mTc]-HMPAO SPECT was performed as a method to detect brain perfusion patterns. The patterns of abnormal brain perfusion cannot always be reliably and qualitatively assessed when dealing with rare diseases. Recent advances in SPECT quantification using commercial software have enabled more objective and detailed analysis of brain perfusion. The cerebellum and whole brain were used as the normalization structures and were compared with visual analysis. Results: The quantification analysis performed with whole brain normalization confirmed right parietal lobe hypoperfusion while also detecting statistically significant left-to-right perfusion differences between the temporal lobe and thalamus. Whole brain normalization further described bilateral frontal lobe hyperperfusion, predominantly of the left lobe, and was in accordance with visual analysis. Conclusion: SPECT quantitative brain perfusion analysis, using the whole brain as the normalization structure rather than the cerebellum, in this case, improved confidence in the visual detection of anti-NMDAR encephalitis and provided unexpected solutions to atypical psychiatric dilemmas.

## 1. Introduction

Anti-N-methyl-D-aspartate receptor (anti-NMDAR) encephalitis is a treatable immune-mediated disease characterized by a complex neuropsychiatric syndrome with autonomic symptoms, commonly misdiagnosed as schizophrenia [1,2]. The incidence in the population is about 0.83 per million, including a female-to-male ratio of 4:1 [3]. Over 90% of patients develop psychiatric symptoms that can be difficult to differentiate from a primary psychiatric disease, primarily schizophrenia and psychotic spectrum disorders [4]. In a number of known cases, a psychiatric assessment is the first step when dealing with a newly discovered illness because the frequency and severity of psychiatric symptoms play a central role in the discovery of this rare disease [5]. More frequent symptoms in adults are psychosis and seizures, abnormal behavior, cognitive and memory impairments, orolingual dyskinesias, chewing movements, hypoventilation, and myoclonus starting with viral— symptoms such as dizziness, fever, nausea, vomiting, headache, abdominal pain, sleeping disorders, and high blood pressure. Autonomic instability, such as variation in blood pressure, temperature, heart rate dysfunction, and cardiac arrhythmias, develop [6]. The main pathophysiological mechanism related to anti-NMDAR encephalitis are antibodies against the GluN1, NR subunit 2A and 2B of NMDAR and are detected in cerebrospinal fluid (CSF) and serum [7]. Antibodies cause a reduction in synaptic clusters of NMDAR’s, which is similar to the NMDAR hypofunction hypothesis of schizophrenia [8]. Neurological disease can be suspected if anti-NMDA receptor antibodies are positive in patients with psychiatric symptoms. A higher level of antibody titer presented with psychiatric symptoms at the beginning of the disease expresses more severe clinical manifestations [9]. Up to 60% of patients with anti-NMDAR encephalitis have underlying tumors, mostly ovarian teratomas, but other tumors such as testicular carcinoma, colon carcinoma, lung carcinoma, thyroid carcinoma, and neuroblastoma have also been reported [10]. Derived from the neural crest, pheochromocytoma is a chromaffin cell tumor associated with catecholamine production, most commonly appearing in the third to fifth decade of life, and can potentially be the trigger of anti-NMDAR encephalitis [11,12,13,14]. In about 40% of patients, the etiology of this disease is non-paraneoplastic, particularly in younger patients. One of the confirmed triggers of anti-NMDAR encephalitis is herpes simplex encephalitis [10,15]. It is important to emphasize that approximately 10% of patients with schizophrenia were positive for anti-NMDA receptor antibodies [16]. Due to the presence of hallucinations, catatonia, and delusions, the primary differential diagnoses of anti-NMDAR encephalitis are psychiatric diagnoses [17].

Diagnostic procedures in a patient presenting with combined neurological and psychiatric symptoms usually include multislice computer tomography of the whole body (MSCT), magnetic resonance imaging (MRI) of the brain, electroencephalography, 18F-fluoro-2-deoxy-D-glucose positron emission tomography (2-[18F] FDG PET), and a neuropsychological assessment. Due to the high incidence of teratoma, transvaginal ultrasound in female patients is obligate. Electroencephalography shows abnormal results in almost all cases but is non-specific. Although abnormal findings may be present in some patients, a pathognomonic sign of anti-NMDAR encephalitis has still not been established. Drug intoxications, lethargic encephalitis, neuroleptic malignant syndrome, or viral encephalitis should be considered in differential diagnosis [6]. Routine blood tests are non-specific. Its rarity, along with the potentially exotic clinical presentation, renders it unlikely for inclusion in differential diagnosis during standard clinical and imaging diagnostic procedures. The timely detection of potential paraneoplastic initiators is important because treatment is more effective if the underlying tumor is removed [18]. The demonstration of IgG antibodies against the GluN1 subunit of the receptor in CSF is the only specific diagnostic test for anti-NMDAR encephalitis [19]. Therefore, only when anti-NMDAR encephalitis is suspected can CSF antibody analysis confirm the diagnosis. Confirmation is crucial because the early identification and treatment of anti-NMDAR encephalitis is a major positive prognostic factor for illness outcomes [20].

Morphological imaging studies, primarily MR, are usually non-specific and mostly without abnormalities at initial presentation [21,22]. Recently published analysis based on voxel-based analysis showed gray matter atrophy and decreased volume of gray matter in bilateral thalamus, frontal and temporal lobes, and sometimes, T2/FLAIR hyperintensity in the hippocampus [23]. Published research on metabolic brain patterns in anti-NMDAR encephalitis by using 2-[18F] FDG PET/CT described mixed metabolic patterns consisting of hypometabolism in the occipital and parietal lobe, as well as hypermetabolism in the temporal lobe and cerebellum [24]. It is also noted in a recent case report study that cerebral hypometabolic patterns were in accordance with receptor hypofunction and that reversibility of hypometabolism correlated with patients’ cognitive improvement [25].

Brain [99mTc]-hexamethyl propylene amine oxime (HMPAO) single-photon emission computed tomography (SPECT) is an alternative method to 2-[18F] FDG PET/CT mainly by being able to detect brain perfusion defects while FDG PET/CT presents the metabolic changes in the brain. Whether talking about perfusion in the case of SPECT studies or metabolism in PET/CT studies, perfusion or metabolic distribution is almost the same. There is still a huge number of nuclear medicine departments without PET/CT devices. The diagnostic abilities of SPECT in comparison with 2-[18F] FDG PET/CT were nearly identical in studies with Alzheimer’s disease and minimal cognitive impairment [26]. Cases of functional SPECT studies in anti-NMDAR encephalitis have been published with contradictory findings. Some reported no significant changes in brain SPECT studies, while a minority of the patients presented with frontotemporal hyperperfusion [27]. Furthermore, other cases reported intense hyperperfusion of basal ganglia, especially on the left side [25].

Difficulties in timely disease detection emphasize the need for a more precise imaging method. Recent advances in SPECT quantification have enabled a more objective and detailed analysis of brain perfusion. The aim of this article is to present the potential use of SPECT quantification in addition to qualitative brain perfusion analysis for the detection of anti-NMDAR encephalitis and to emphasize the importance of quantification in SPECT studies. 

## 2. Case Presentation

A 58-year-old sailor was hospitalized in our institution for a few months after coming back from Lagos (Africa). His wife noticed sleeping and speech disorders lasting up to two months, spatial disorientation, frequent falls, hallucinations, and confused and disturbed thoughts. Furthermore, driving and social abilities were reduced. These symptoms were accompanied by delusions, food refusal, progressive cognitive decline, involuntary jerks of his left hand, weakness, and balance disturbance. Psychiatric and psychological examinations were performed. Psychiatric status at admission was: Conscious, orientated adequately in all directions apart from time, contact directed by assistance. Psychomotor retardation was evident, with reduced mood and flat affect. The thought processes were coherent, with suspected paranoid delusions but not suicidal ideation. The Mental State Examination (MMSE) score was 29/30, while the Positive and Negative Syndrome Scale (PANSS) score was 114, dominated by negative symptomatology (score of 35). The psychiatrist made the diagnosis of an organic delusional (schizophrenia-like) disorder—F06-2, according to ICD-10. Lorazepam (1.25 mg orally twice a day) and haloperidol (2 mg orally trice a day) were started, later augmented with quetiapine (25 mg orally trice a day). The results of the psychological testing indicated specific cognitive deviations similar to those usually established in organic mental disorders: significant disturbances of attention and concentration, deficits in the ability to adopt new verbal content, complete loss of ability to adopt new nonverbal content, visual–motor impairment, and significant psychomotor retardation. Due to the patient’s lack of insight and denial, it was not possible to apply objective personality questionnaires during the examination. The examiner also noted indicators of considerable regression, as well as increased sensitivity to others’ opinions. Frequent myoclonus occurred, dominant on the left side, with one grand mal seizure documented during the hospitalization. All of the infective causes of these symptoms were ruled out. Extensive laboratory testing for autoimmune, hypercoagulable, and paraneoplastic diseases was performed. Brain MSCT with intravenous contrast revealed only mild atrophic changes, while Gadolinium-enhanced brain MRI revealed chronic vascular white matter lesions, mostly right-sided, along with changes in the posterior part of the corpus callosum that suggested a metabolic or toxic etiology of the disease. The visual analysis of the brain perfusion SPECT with [99mTc]- HMPAO revealed dominant right parietal and medial temporal hypoperfusion, along with the hypoperfusion of the right thalamus and striatum. Furthermore, minor hypoperfusion of the left frontal lobe was described. The subsequent analysis of the S-anti-NMDAR and L-anti-NMDAR antibodies in the blood and CSF were positive, suggesting the diagnosis of anti-NMDAR encephalitis. Despite plasma exchange and immunoglobulin immunomodulatory therapies, which are the usual therapies for anti-NMDAR encephalitis, the patient deteriorated mnestically, cognitively, and neurologically. Searching for the etiology of the disease 2-[18F]FDG PET/CT was performed. Based on the PET/CT findings, as well as the clinical presentation, a right adrenal gland tumor was found. Surgery was performed immediately, and the results of the pathohistological analysis confirmed the diagnosis of pheochromocytoma. During the two-year follow-up period after surgery, the patient’s verbal and nonverbal intellectual abilities, as well as verbal fluency, visuomotor coordination, and psychomotor speed, improved. Furthermore, the psychiatric symptoms resolved, and psychiatric therapy was gradually reduced. Deficits of visual memory and visuoconstructive spatial abilities were still pronounced, with working memory and the ability to adopt new content being significantly reduced. Involuntary jerks of his left hand, weakness, and balance disturbance also subsided. 

### 2.1. Methods

A brain perfusion SPECT study using [99mTc]-HMPAO was performed. The patient was injected with 1100 MBq of [99mTc]-HMPAO (National Centre for Nuclear Research Radioisotope Centre, Polatom, Otwock-Swierk, Poland) in its unstabilized form with a radiochemical purity greater than 90%. SPECT was acquired using a double-headed rotating gamma camera (Symbia T Series SPECT/CT—Siemens Healthcare GmbH, Germany), twenty minutes after injection, with a parallel-hole, low-energy high-resolution collimator. The acquisition mode was a step and shot, 32 projections, 40 s per view, matrix size 128 × 128. Filtered back projection reconstruction and Chang’s attenuation correction were performed. After visual analysis, software analysis using commercial software was made. (MIM neuro, Version 7.1.6, 25,800 Science Park Drive Suite 180 Cleveland, OH 44122, USA).

The regions of interest were generated using the brain atlas template: the frontal, parietal, temporal, and occipital lobes, medial and the lateral temporal lobes, superior parietal lobe, cerebellum, and thalamus were analyzed. Autorecognition for brain regions was used and was corrected by nuclear medicine specialists, as displayed in Figure 1. 

To perform a quantitative assessment, the analysis was repeated two times, first by using the whole brain and then by using the cerebellum as a normalization structure for the relative brain activity. The selected patient’s brain structures were then compared with multiple age-matched studies available from a database of healthy patients. Each region was presented with a Z-score, the Z-score for the left and right sides, and L–R % difference Z-score. The threshold value for the Z-score was set to 1.65 to detect statistically significant results in accordance with the software performance. Two nuclear medicine specialists interpreted the SPECT images using a visual qualitative approach and then using a described quantitative software-aided approach. The data provided by the quantitative SPECT analysis are presented in Table 1 and are displayed as Z-score values in regard to brain perfusion in the control group of healthy age-matched patients. 

The study was conducted in accordance with ethical standards set by the institutional Ethics Committee and the Helsinki Declaration from 1975, as revised in 2013 [28]. Ethical review and approval were waived due to the nature of this study as the performed diagnostic procedure was part of standard diagnostic procedures, and the subsequent quantitative software analysis presented no additional risk or intervention to the subject whose medical data were included.

### 2.2. Results

The visual analysis determined segmental hypoperfusion, including the right parietal lobe, right temporal lobe, as well as the right thalamus and frontal lobe. The reconstructed SPECT study is presented in Figure 2, and is displayed as standard transverse projection, which is used in routine clinical practice. 

In comparison, the brain quantification analysis performed with cerebellum normalization detected statistically significant left-to-right difference in described regions, but without true perfusion deficit and also presenting with left frontal hyperperfusion. Further performed quantification analysis with whole brain normalization confirmed right parietal lobe hypoperfusion while also detecting statistically significant left-to-right perfusion difference of temporal lobe and thalamus. Whole brain normalization further described bilateral frontal lobe hyperperfusion, predominantly of the left lobe. The data provided by the quantitative SPECT analysis is presented in Table 2 and compared with the visually detected perfusion anomalies.

Quantitative analysis is further visually displayed in Figure 3 and represents the visual cortex mapping of the brain in regard to the detected Z-score values in comparison to the database of normal HMPAO SPECT perfusion studies. 

## 3. Discussion

The quantitative analysis of brain perfusion scans can be very helpful as an aid to visual analysis in challenging clinical cases. Nuclear medicine neurology is primarily based on PET/CT studies, which natively support quantification analysis as a comparative advantage to standard SPECT/CT studies. There are a few approaches for intensity normalization in PET/CT, such as predefined, spared, or reference region means of the average uptake. Across the different studies, different target regions were also investigated, including pons, cerebellum, brainstem or primary sensorimotor cortex, and the whole brain in different commercially available software packages, such as Statistical Parametric Mapping (SPM, Wellcome Department of Imaging Neuroscience, London, United Kingdom) [29,30,31]. However, PET mainly presents the distribution of brain metabolic activity compared to the brain perfusion detected by HMPAO SPECT. The patterns of abnormal brain perfusion cannot always be reliably and qualitatively assessed when dealing with rare diseases. The advantages of SPECT quantification methods are that they provide a means to perform pattern recognition in problematic clinical cases better. To be able to compare the patient results with the group of age-matched healthy patients objectively, a SPECT perfusion study must first be mapped, and we used voxel-to-voxel mapping as the standard brain template. The question is how to normalize brain activity to be able to quantitatively detect the perfusion patterns. Usually, brain activity is normalized to a structure that is considered unaffected by the disease. Recent research in other neurological diseases has shown that, for example, Alzheimer’s disease scans should be normalized to the cerebellum as it is preserved during disease [32]. Anti-NMDAR encephalitis currently does not have a brain region proven to be spared during disease onset, and as discovered by PET, the cerebellum can also display some kind of metabolic change, in this case, hypermetabolism [24]. While using the cerebellum as the normalization structure in our analysis, we would have lower concordance with results described in published cases. Furthermore, the visual analysis does not match the quantitively assessed analysis. However, normalizing to the whole brain presents with the pattern visually described in multiple cases thus far, and these quantitative results are beneficial when the interpreter is not familiar with anti-NMDAR encephalitis qualitative presentation. We have quantitatively determined significant parietal hypoperfusion, statistically significant perfusion asymmetry in the temporal lobe and thalamus together with the bilateral hyperperfusion of the frontal lobes, comparable with part of visually detected perfusion abnormalities in earlier described studies [27,33,34,35,36]. In contrast, we have not detected abnormal perfusion in the cerebellum or hyperperfusion of temporal and occipital lobes [34,36,37]. As an advantage of quantitative analysis, we noticed that visual analysis can recognize an asymmetry between the left and right lobe, or any subsegmental part, but cannot visually presume whether this is the better-perfused side normal or hyperperfunded, which can be the major dilemma in different psychiatric or encephalitis related states. Furthermore, by comparing visual and quantitative analysis, using quantitative analysis, it can be measured whether the difference between opposite sites is objective and statistically significant in comparison with a database of healthy patients. Using sophisticated quantification software, we can compare the region of interest with multiple age-matched normal patient databases or construct a comparative database of similar clinical cases. Moreover, it is possible to choose a normalization structure depending on the research target or sparing area in the particular disease. Both hypoperfusion and hyperperfusion Z-scores can be calculated, offering the possibility of clinical monitoring and the influence of therapy on the outcome of the disease. 

## 4. Conclusions

This case presents a potentially beneficial approach for detecting a rare disease and is an example of SPECT quantification analysis as a missing step in resolving clinical questions. It is limited because it only presents the results of only one patient. However, to our knowledge, it is the only known published case of anti-NMDAR encephalitis caused by pheochromocytoma evaluated with brain perfusion single-photon tomography. It is important to emphasize that while SPECT study analyses have been performed and published, a quantitative comparison of this type has not been described in cases of anti-NMDAR encephalitis. Further studies with quantitative analysis of multiple affected patients would provide a basis for determining specific [99mTc]-HMPAO SPECT brain perfusion patterns. We conclude that SPECT quantitative brain perfusion analysis using the whole brain as a normalization structure improved confidence in visual detection of anti-NMDAR encephalitis and provided unexpected solutions to atypical psychiatric dilemmas. Furthermore, adding quantification in routine clinic use with SPECT/CT can provide more convincing results in departments without PET/CT equipment and can provide better treatment results.

## Figures and Tables

**Figure 1 diagnostics-12-02377-f001:**
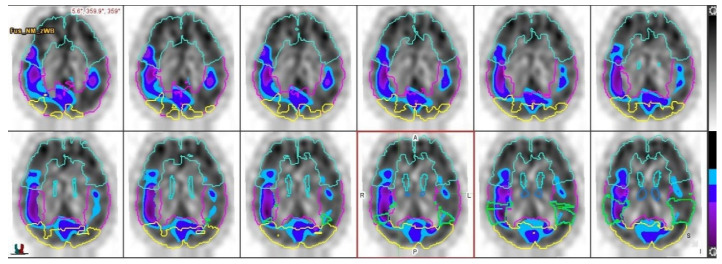
Presentation of performed autodetection of selected brain regions on SPECT transverse view, based on brain atlas template.

**Figure 2 diagnostics-12-02377-f002:**
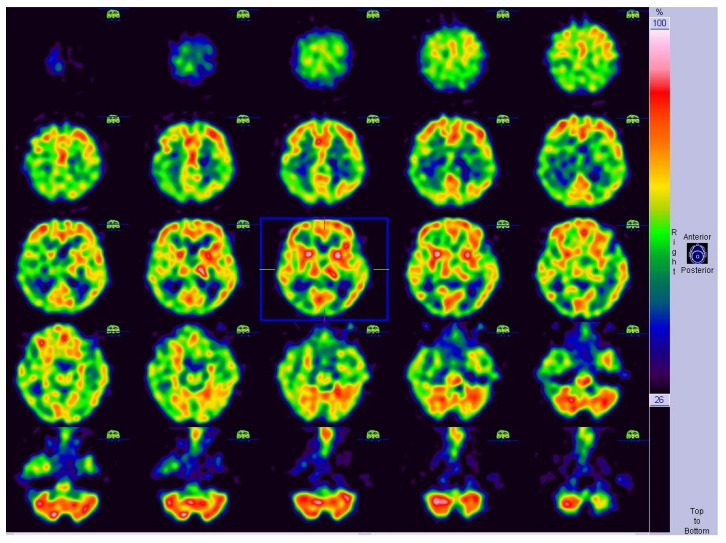
Reconstructed 99mTc-HMPAO SPECT of the brain presented in transverse view and used for qualitative analysis in described case.

**Figure 3 diagnostics-12-02377-f003:**
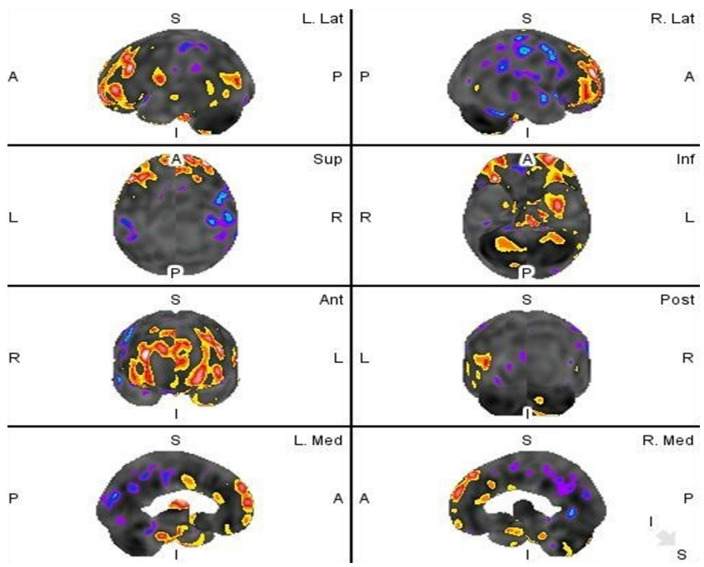
Two-tail Z score presentation of regions with statistically significant hypoperfusion (blue) and hyperperfusion (red).

**Table 1 diagnostics-12-02377-t001:** Z-score results from SPECT quantitative analysis, B—perfusion normalized to whole brain C- perfusion normalized to cerebellum.

Region	Z-Score	Z-Score	LZ-Score	LZ-Score	RZ-Score	RZ-Score	L-R%diff	L-R%diff	L-R% Diff Z-Score	L-R% Diff Z-Score
	B	C	B	C	B	C	B	C	B	C
Parietal lobe	−1.82	−0.69	−0.99	−0.22	−2.4	−1.16	4.7	4.7	2.19	2.19
Superior parietal lobe	−0.59	−0.15	0	0.25	−1.1	−0.62	6.12	6.12	1.78	1.78
medial temporal lobe	−0.03	0.44	0.94	1.05	−0.82	−0.14	3.66	3.66	1.37	1.37
Thalamus	−0.34	0.62	1.13	1.17	−0.62	−0.3	5.54	5.54	1.81	1.81
Occipital lobe	−0.26	0.23	0.44	0.61	−0.84	−0.1	6.17	6.17	1.61	1.61
Temporal lobe	0.55	0.62	2.37	1.43	−1.46	−0.21	5.62	5.62	2.64	2.64
Lateral temporal lobe	0.57	0.63	1.97	1.3	−0.88	−0.8	5.62	5.62	2.07	2.07
cerebellar hemisphere	−0.53	X	0.3	X	−1.18	X	3.04	X	2.25	X
frontal lobe	3.97	1.44	4.2	2.07	1.45	0.85	3.62	3.62	2.35	2.35

**Table 2 diagnostics-12-02377-t002:** Comparison of visualy detected perfusion anomalies with quantitatively detected perfusion changes.

Visual Analysis/Hypoperfusion Areas	MiM Software Quantification/Whole Brain Normalization	MiM Software Quantification/Cerebellum Normalization
Parietal lobe (right)	Confirmed, right parietal lobe hyperperfusion	Significant statistical difference in perfusion between L and R, but without true right parietal hypoperfusion
Temporal lobe (right)	Significant statistical difference in perfusion between L and R, but without true right temporal hypoperfusion	Significant statistical difference in perfusion between L and R, but without true right temporal hypoperfusion
Thalamus (right)	Significant statistical difference in perfusion between L and R, but without true right thalamus hypoperfusion	Significant statistical difference in perfusion between L and R, but without true right thalamus hypoperfusion
Frontal lobe (right)	Bilateral frontal lobe hyperperfusion, primarily left frontal lobe	Left frontal lobe hyperperfusion

## Data Availability

The data presented in this study are available upon request from the corresponding author. The data are not publicly available due to the GDPR privacy policy.

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
