# Peer review of "Psychiatric Illness or Immune Dysfunction—Brain Perfusion Imaging Providing the Answer in a Case of Anti-NMDAR Encephalitis"

_diagnostics, 2022, doi:10.3390/diagnostics12102377_

Round 1
Reviewer 1 Report
The study, “Psychiatric illness or Immune dysfunction – Brain perfusion imaging providing the answer in a case of anti-NMDAR encephalitis”, by Ines Sisko Markos, el al. has described an interesting SPECT quantitative brain perfusion analysis for detection of anti-NMDAR encephalitis. The study shows the case of a male patient and the objective and detailed analysis of brain perfusion. It could be valuable study to be published in Diagnostics with minor revisions.
- For decimal numbers, comma(,) or point(.) should be used in the MDPI journal?
- (Page 7, 205 line) Can you add the reference?
- In the table 2, the words ‘hyperperfusion’ are italic at the last row only. Is there any meaning?
- Figure 3 seems cut the right part of the figure.
Author Response
Dear reviewer,
Thank you for your time and efforts made in reviewing our manuscript and thank you for your valuable suggestion.
Point 1: For decimal numbers, comma(,) or point(.) should be used in the MDPI journal?
Response 1: No strict direction about comma or point, but now all text is uniform.
Point 2: (Page 7, 205 line) Can you add the reference?
Response 2: We added a reference.
Point 3: In the table 2, the words ‘hyperperfusion’ are italic at the last row only. Is there any meaning?
Response 3: No meaning, lapsus.
Point 4: Figure 3 seems cut the right part of the figure.
Response 4: The whole figure was present in manuscript but Word form reformatted it out of borders. Figure is now correctly aligned.
Reviewer 2 Report
This case presented limited results in one patient.
However, the SPECT image of NMDAR encephalitis is very interesting and, if published, may be helpful to readers.
I suggest changing the format of the paper to interesting images and submitting it.
We thank the authors for their hard work.
Author Response
Point 1.
This case presented limited results in one patient. However, the SPECT image of NMDAR encephalitis is very interesting and, if published, may be helpful to readers. I suggest changing the format of the paper to interesting images and submitting it.
Response 1.
Dear reviewer,
Thank you for your time and efforts made in reviewing our manuscript and thank you for your valuable suggestion. During our initial submission we submitted the manuscript in a shorter format, but the Editor suggested to expand the content of our manuscript which we did. In our opinion the expanded manuscript is more informative and of higher value for readers because of the better comparison made in the manuscript with other studies and we would prefer not to again reduce it to interesting image format.
Round 2
Reviewer 2 Report
The paper has been well edited.